# Titanium Surface Modification for Implantable Medical Devices with Anti-Bacterial Adhesion Properties

**DOI:** 10.3390/ma15093283

**Published:** 2022-05-03

**Authors:** Consuelo Celesti, Teresa Gervasi, Nicola Cicero, Salvatore Vincenzo Giofrè, Claudia Espro, Elpida Piperopoulos, Bartolo Gabriele, Raffaella Mancuso, Giovanna Lo Vecchio, Daniela Iannazzo

**Affiliations:** 1Department of Engineering, University of Messina, Contrada Di Dio, I-98166 Messina, Italy; ccelesti@unime.it (C.C.); espro@unime.it (C.E.); epiperopoulos@unime.it (E.P.); 2Department of Biomedical and Dental Sciences and Morphological and Functional Images, University Hospital of Messina, Via Consolare Valeria, 1, I-98100 Messina, Italy; teresa.gervasi@unime.it (T.G.); nicola.cicero@unime.it (N.C.); 3Science4Life srl, Spin-off Company, University of Messina Viale Ferdinando Stagno D’Alcontres, 31, I-98166 Messina, Italy; 4Department of Chemical, Biological, Pharmaceutical and Environmental Sciences, University of Messina, Viale Annunziata, I-98168 Messina, Italy; salvatorevincenzo.giofre@unime.it (S.V.G.); giovanna.lovecchio@unime.it (G.L.V.); 5Laboratory of Industrial and Synthetic Organic Chemistry (LISOC), Department of Chemistry and Chemical Technologies, University of Calabria, Via Pietro Bucci 12/C, I-87036 Arcavacata di Rende, Italy; bartolo.gabriele@unical.it (B.G.); raffaella.mancuso@unical.it (R.M.)

**Keywords:** metal surface modification, anti-bacterial adhesion, titanium implants

## Abstract

Pure titanium and titanium alloys are widely used in dentistry and orthopedics. However, despite their outstanding mechanical and biological properties, implant failure mainly due to post-operative infection still remains a significant concern. The possibility to develop inherent antibacterial medical devices was here investigated by covalently inserting bioactive ammonium salts onto the surface of titanium metal substrates. Titanium discs have been functionalized with quaternary ammonium salts (QASs) and with oleic acid (OA), affording the Ti-AEMAC Ti-GTMAC, Ti-AUTEAB, and Ti-OA samples, which were characterized by ATR-FTIR and SEM-EDX analyses and investigated for the roughness and hydrophilic behavior. The chemical modifications were shown to deeply affect the surface properties of the metal substrates and, as a consequence, their bio-interaction. The bacterial adhesion tests against the Gram-negative *Escherichia Coli* and Gram-positive *Staphylococcus aureus*, at 1.5 and 24 h of bacterial contact, showed good anti-adhesion activity for Ti-AUTEAB and Ti-OA samples, containing a long alkyl chain between the silicon atom and the ammonium functionality. In particular, the Ti-AUTEAB sample showed inhibition of bacteria adhesion against *Escherichia Coli* of about one log with respect to the other samples, after 1.5 h. The results of this study highlight the importance of chemical functionalization in addressing the antimicrobial activity of metal surfaces and could open new perspectives in the development of inherent antibacterial medical devices.

## 1. Introduction

Wearable and implantable medical devices play a fundamental role in modern healthcare. The implant technology has dramatically improved the quality of life of millions of people worldwide, allowing the integration in the human body of different medical solutions such as reconstructive joint replacements, programmable pacemakers, defibrillators, stents, synthetic valves, tissue-engineered scaffolds, and drug delivery systems [1]. Even if medical devices constituted by ceramics and polymers have shown excellent biofunctionality and biocompatibility, more than 70% of surgical implants are still constituted by metal-based implants [2,3]. This preference is mainly due to their durability and high fracture toughness. Among the different metallic materials used for medical implants, pure titanium and titanium alloys are the most used in dentistry and orthopedics because of their unique mechanical properties, corrosion resistance in biological media, and their good biocompatibility and osseointegration ability [4,5,6]. However, despite the outstanding mechanical and biological properties of titanium-based devices, the implant failure (mainly due to postoperative infection and/or immune responses) still remains a significant concern, leading to adverse clinical outcomes including bone loss and, in the most severe cases, even death [7,8]. The primary causes of medical device complications, namely infection and inflammation, are mediated by the interaction of the implanted materials with the host environment. In fact, the implanted devices could be contaminated before implantation with bacteria from droplets, air, or hands of healthcare workers or immediately upon their introduction. After their introduction into the human body, these devices can be coated with serum proteins, aqueous humor, mucosal secretions, host microbiota, and extracellular fluids. These events make the materials’ surface highly susceptible to microbial infection and host cellular adhesion [9,10]. Pathogens thus anchored on the titanium surface through various surface interactions like van der Waals forces, hydrogen bonding, hydrophobic interactions, and ionic interactions, are able to multiply and colonize the surface, consequently resulting in device infection [11]. Therefore, inhibiting bacterial colonization is of pivotal importance to avoid the probability of infection.

Several strategies have been proposed to overcome these serious drawbacks, with preventive strategies being the most effective. In this context, the use of intrinsic antimicrobial surfaces represents the most advantageous possibility [12,13]. In general, two strategies have been investigated to inhibit bacterial colonization on surfaces: (i) coating the device surface with an antifouling layer to prevent bacterial adhesion, or (ii) coating the device surface with a bactericidal layer [14]. The first approach is based on the formation of a hydrated barrier able to inhibit a bacterial attack [15,16]. This strategy can retard biofilm formation but does not lead to the killing of bacteria cells that are already deposited on the surface [17]. The second approach is based on the covalent immobilization of bactericidal agents, like antibiotics, antiseptics, and silver or nitric oxide on the surface of the implant [18,19]. This second strategy shows some disadvantages, such as the low concentration of loaded bactericidal agents, a fast release over time into body fluids with consequent toxicity profiles, problems related to antibiotic resistance, and above all, the poor efficiency for long periods [20,21]. Additionally, the use of metal coatings which are extremely effective in preventing biofilm formation can lead to severe toxicity problems due to their accumulation in tissues which causes long-term damage [22]. In this scenario, the covalent anchoring of biomolecules with antibacterial properties, such as quaternary ammonium compounds and antimicrobial peptides on medical implants has shown to afford new materials able to kill microorganisms on contact, thus exhibiting a self-sterilizing effect [13,23,24]. This strategy proved to overcome problems related to biofilm-associated infections in medical devices with minimal cytotoxicity and stability profiles [24,25,26,27]. 

Among the different antibacterial compounds, quaternary ammonium salts (QASs) exert a strong antimicrobial activity against a wide class of microorganisms including bacteria, fungi, and viruses [28,29]. Their permanent positive charge allows a fast binding with the negatively charged surface of most microbes, thus leading to the loss of cell membrane integrity and then cell death. Based on these considerations, the aim of this work was to investigate for the first time the possibility of covalently inserting bioactive QASs moieties directly onto the surface of titanium metal substrates for the development of inherent antibacterial medical devices. The QASs chosen in this study were: 2-(acryloyloxy)ethyl]trimethylammoniumchloride (AEMAC), glycidyltrimethylammonium chloride (GTMAC), and acryloyloxyundecyltriethylammonium bromide (AUTEAB). The latter, previously synthesized by us [30,31], proved to be a good antibacterial agent [32,33]. 

The QASs were covalently anchored to the surface of titanium discs by means of the linker (3-aminopropyl)triethoxysilane (APTES). The alkoxy groups of this molecule were linked to the titanium hydroxyl groups by siloxane bonds while the nucleophilic amino group was used for the further bonding with QASs. Moreover, we also investigated the antimicrobial activity of ammonium salt functionalized titanium discs obtained by reaction of the naturally-derived oleic acid (OA) with the organosilane (*N*,*N*-dimethylaminopropyl)trimethoxysilane (DAPTES); analogously to APTES, also DAPTES was anchored to the titanium surface by siloxane bonds, while the tertiary amino group was used to form the ammonium salt with the carboxyl group of OA.

The so functionalized titanium discs were evaluated for their bacterial adhesion inhibition against the Gram-negative *Escherichia Coli* and Gram-positive *Staphylococcus aureus*, showing a clear anti-adhesion activity for Ti-AUTEAB and Ti-OA samples, containing a long alkyl chain between the silicon atom and the ammonium functionality. The results of biological preliminary tests highlight the importance of chemical functionalization in addressing the antimicrobial activity of metal surfaces and open new perspectives in the development of antimicrobial implantable titanium medical devices.

## 2. Materials and Methods

### 2.1. Chemicals and Instrumentation

Commercial titanium discs (25 mm in diameter and 0.5 mm thick) were purchased from Merck Life Science with chemical and mechanical properties according to the standard ASTM-B-265/ASME-SB-265. Oleic acid, 3-aminopropyltriethoxysilane, *N,N*-dimethylaminopropyl-trimethoxysilane, sodium hydroxide (pellets), and the solvents toluene (99.8%) ethanol (99.8%) and methanol (99.8%), were obtained from Merck Life Science at the highest purity available and used without further purification. Acryloyloxyundecyltriethylammonium bromide (AUTEAB) was prepared as we already reported [30,31]. Spectrometer (PerkinElmer, Waltham, MA, USA), by the method ATR in the range of 4000–500 cm^−1^. Scanning Electron Microscopy (SEM) was conducted at room temperature with an FEI Quanta 450 FEG instrument (Thermo Fisher Scientific, Hillsboro, OR, USA) operating in a high vacuum, at 20 kV, using an Everhart-Thornley detector (ETD). The Energy Dispersive X-ray (EDX) analyses were made using an Octane Plus Silicon Drift Detector (Ametek, Berwyn, PA, USA), equipped with a 30 mm^2^ Super Ultra Thin Window (SUTW). The FeK mapping analysis was performed using an image resolution of 256 × 200 pixels and a dwell time of 200 µs. The mapping acquisition time was set at 60 min. The roughness of the samples was measured using a Surftest SJ-210 roughness meter Series 178 (MitutoyoS.r.l., Milan, Italy). In particular, the arithmetic average height (Ra) [µm] was evaluated by:(1)$$Ra=\frac{1}{N}\overset{n}{\underset{i=1}{\sum }}\left|Yi\right|$$
where *Ra* represents the arithmetic mean of the absolute values of the evaluation profile (*Yi*) from the mean line. The measurement conditions were regulated according to the JIS2001 roughness standard, five sampling lengths, lengths of cut-off (λs = 2.5 μm, λc = 0.8 μm), and a stylus translation speed of 0.5 mm/s. Four roughness profiles per type of sample were performed and then an average profile was obtained. Contact angle analysis was performed using the static method and direct measurement of the tangent angle at the three-phase contact point on a sessile drop profile, using high-resolution photographs of pure water droplets and graphic image processing software (Gimp). In particular, five sets of contact angle measurements on each sample, were performed. The volume of the droplets was 1 μL. On each set we measured the contact angle from both sides (left and right), each reading was taken 5 times. The following results present the average of these measurements.
(2)$$\theta_{w}=2\mathrm{arctg}\left(2h/d\right)$$
where *d* is the diameter and h is the height (both in mm) of the drop, θ_w_ is the apparent Wenzel angle, dependent on the roughness (r) of the surface 

### 2.2. Functionalization of Titanium-Based Samples

#### 2.2.1. Activation of Titanium Samples

Titanium discs were cleaned by ultrasonication in acetone, ethanol, and deionized water (10 min each). The surfaces of 15 titanium discs were then activated by immersing the discs for 48 h into a 5 M NaOH solution (100 mL) and heating at 60 °C in a glass flask fitted with a reflux condenser. After this treatment, the samples were washed with ethanol and dried with nitrogen gas. These alkali-activated samples were labeled as Ti-OH.

#### 2.2.2. Silanization of Activated Titanium Discs

Ti-OH samples (14 discs) were subjected to silanization using solutions (2.0 vol.%) of APTES or DAPTES in anhydrous toluene. The samples, placed in a glass flask fitted with a reflux condenser were heated for 36 h at 70 °C, under a nitrogen atmosphere. Afterward, the discs were profusely washed with ethanol and dried with nitrogen. They were respectively labeled as Ti-APTES (6 discs) or Ti-DAPTES (6 discs). 

#### 2.2.3. Synthesis of AEMAC, AUTEAB, and GTAMC Functionalized Titanium Discs 

Ti-APTES samples (3 discs) were placed in a flask fitted with a reflux condenser and then treated with a solution of AEMAC (0.28 mmol), AUTEAB (0.28 mmol), or GTMAC (0.28 mmol), in methanol (25 mL). The disks were heated for 36 h at 80 °C and then washed with ethanol and dried with nitrogen. The as-prepared samples were named Ti-AEMAC, Ti-AUTEAB, or Ti-GTMAC, respectively.

#### 2.2.4. Synthesis of Oleic Acid Salts Functionalized Titanium Disks

Ti-DAPTES samples (3 discs), placed in a flask fitted with a reflux condenser were treated with a solution of oleic acid (OA, 0.017 mmol), in methanol (10 mL). The discs were heated for 36 h at 80 °C and then were washed with ethanol and dried with nitrogen. The as-prepared samples were named Ti-OA. 

### 2.3. Evaluation of Titanium Discs’ Anti-Bacterial Adhesion Properties

#### 2.3.1. Microbial Strains and Culture Conditions

The following American Type Culture Collection (ATCC) strains were used for the adhesion testing: *Escherichia coli* ATCC 25922 and *Staphylococcus aureus* ATCC6538. For the susceptibility studies, the strains were grown in Mueller Hilton Broth (MHB, Oxoid, CM0405, Sigma, Milan, Italy) at 37 °C for 18–20 h. 

#### 2.3.2. Adhesion Tests

The bacterial adhesion on titanium discs’ surfaces was performed on bare titanium discs and on the functionalized samples as reported in the literature [32]. Briefly, the overnight cultures were harvested by centrifugation (3000× *g* for 15 min), washed twice using phosphate-buffered saline (PBS) solution, and suspended to a concentration of 1 × 10^6^ Colony Forming Unit (CFU)/mL in PBS solution. The titanium discs, which were sterilized in an ethanol bath rinsed in sterile distilled water, and dried under a sterile atmosphere, were placed in sterile Petri dishes, covered with 1 ml of bacterial suspension (10^6^ CFU/mL), and incubated at room temperature for 1.5 h and 24 h. Then, in order to remove adhering bacteria, samples were rinsed with sterile distilled water and sonicated for 3 min in 5 mL of sterile PBS solution. Serial dilutions of the PBS solution were performed and were plated in Mueller Hilton Agar (MHA, Oxoid, PO0152, Sigma, Milan, Italy). After 24 h at 37 °C, the colonies were enumerated and expressed as CFU *per* disc (CFU/discs). All experiments were performed in triplicate on three independent days.

#### 2.3.3. Statistical Analysis

The mean differences and standard deviations of antibacterial activity of functionalized titanium discs were calculated with Prism 8.0.2 (GraphPad Software, Inc., La Jolla, CA, USA). Data were first verified with the Shapiro Wilk test for the normality of the distribution and the Brown Forsythe test for the homogeneity of variances. Data were normally distributed and homogenous. Therefore, they were statistically analyzed using one-way analysis of variance (ANOVA) and Tukey post hoc test for multiple comparisons at a level of significance set at *p* < 0.05.

## 3. Results and Discussion

### 3.1. Titanium Discs Activation

The synthetic strategy used to develop inherent antibacterial titanium metal substrates is based on the silanization of titanium surface to immobilize the bioactive molecules. Silane chemistry is a well-established strategy, to chemically modify titanium surfaces, allowing the covalent bonding of biologically active molecules [34]. In particular, the amino silane APTES is a bifunctional molecule, widely used for bioconjugation purposes [35]; the three alkoxy groups of this molecule can be linked to the titanium hydroxyl groups by siloxane bonds while the nucleophilic amino group can be used for further bonding with biologically active compounds. In a similar way, the silane DAPTES can be anchored to the titanium surface by siloxane bonds and the tertiary amino group can react with biologically active compounds [36]. In this study, we used APTES for its ability to readily react with α,β−unsaturated carbonyl compounds or epoxide functionalities and DAPTES, which is able to form ammonium salts after reaction with carboxyl groups (Figure 1). 

To ensure the optimal silanization, the titanium discs were activated by alkaline treatment with NaOH at 60 °C, for 48 h. This increased the number of available hydroxyl groups as confirmed by FTIR spectra (Figure 2). Compared to the untreated sample (Ti), the Ti-OH sample shows a broad peak at 3280 cm^−1^, due to the vibration of the O–H groups and a band centered at 850 cm^−1^, which is normally attributed to the Ti–O bonds [37]. The peak at 1648 cm^−1^ can be related to the presence of adsorbed water molecules on the activated titanium surfaces. The subsequent treatment of Ti-OH with the organosilanes APTES or DAPTES, performed in anhydrous toluene by heating the discs at 70 °C for 36 h, under an inert atmosphere, afforded the silanized discs Ti-APTES and Ti-DAPTES. The FTIR spectra of these samples show the representative peaks located around 1030 cm^−1^, attributable to the presence of the Si–O bond and the aliphatic C–H stretching of the propyl group at 2950–2954 cm^−1^, thus indicating that the silane was covalently bound on the surface of the titanium substrate. Moreover, these samples show the presence of a peak at 3265 cm^−1^, due to the symmetric and asymmetric –NH vibrations and the presence of the peaks at 1580 cm^−1^ and 1640 cm^−1^, ascribable to the bending of N-H group of Ti-APTES and Ti-DAPTES, respectively. Compared to the Ti-OH samples, a weaker broad peak at 3278 cm^−1^ can be observed for both samples, as a consequence of the silanization procedures. 

The activation of the titanium surface by alkaline treatment and the occurred silanization was also confirmed by field emission scanning electron microscopy (FEG-SEM) (Figure 3). As reported in other studies, the chemical treatment with NaOH 5M, in addition to improving the biointegration properties of titanium surfaces [38], was shown to markedly modify the discs’ surface, leading to the almost disappearance of the grinding marks present in the starting substrates. The formation of an amorphous alkali titanate layer was confirmed by EDX spectra, which showed for the oxidized sample, the presence of sodium and oxygen peaks. Elemental analyses via EDX showed average atomic percentages of sodium and oxygen content of 7.4% and 50.7%, respectively. The subsequent silanization procedure revealed for both samples the presence of ordered silicon atoms covering regularly the surfaces of titanium discs with average percentage of silicon atoms of 2% for both silanes. Mapping analyses performed on the silanized samples Ti-APTES and Ti-DAPTES confirmed the presence of homogeneously distributed silicon atoms (yellow spots) covering the surfaces of titanium discs (Figure 4).

### 3.2. Titanium Discs Functionalization

The silanized samples were functionalized with the bioactive molecules to obtain inherently antibacterial titanium metal surfaces. The QASs here investigated, namely GTMAC, AEMAC, and AUTEAB were covalently linked to the Ti-APTES sample by reaction of the free nucleophilic amino group present in the metal substrates with the reactive groups present in the QASs. In particular, AEMAC and AUTEAB were linked to the discs by the Michael addition reaction between the amino group and the terminal double bond present in the QASs while GTMAC was anchored to the surface of the same sample by ring-opening reaction of the epoxide functionality present in GTMAC and the amino group of Ti-APTES. The halogen-free Ti-OA sample, containing the naturally derived oleic acid (OA) was prepared by a simple reaction of the carboxyl functionality present in OA with the dimethylamino functionality exposed on the TI-DAPTES sample, to give the corresponding ammonium salt (Figure 5). 

The effectiveness of functionalization reactions was confirmed by FTIR analyses (Figure 6 and Figure 7). The functionalized samples deriving from Ti-APTES and possessing a quaternary ammonium functionality show a broadband at 3000–2800 cm^−1^ ascribable to the stretching of the N–H bond of the introduced amine salt together with a more intense peak, with respect to the Ti-APTES sample, at 2970–2960 cm^−1^, due to the C–H stretching of alkane functionalities (Figure 6). The Ti-GTMAC sample shows the additional peaks at 2750 cm^−1^ and at 1390 cm^−1^ attributable to the O-H stretching and O-H bending respectively, of alcohol functionality, present in this sample after ring-opening of the epoxide group. The Ti-AEMAC and the Ti-AUTEAB T samples show the presence of a peak at 1735 cm^−1^ and at 1730 cm^−1^ respectively, due to the C=O bond of ester functionalities. Moreover, these samples present the peaks at 1135 cm^−1^ for Ti-AEMAC and at 1130 cm^−1^ for Ti-AUTEAB due to the stretching of C-O groups. The Ti-OA sample, compared with the precursor Ti-DAPTES, shows more intense peaks at 2850 cm^−1^ and 2920 cm^−1^, due to the C–H stretching of newly introduced alkane functionalities. Moreover, for the functionalized sample, additional peaks at 1740 cm^−1^ and 1640 cm^−1^ ascribable to the C=O stretching of carboxylate functionality and to the stretching of the C=C *Z* group respectively, were also recorded (Figure 7). 

### 3.3. Roughness and Contact Angle Measurements 

The chemical functionalization of titanium substrates has been shown to modify the surface roughness and, as a consequence, the interaction of the metal substrate with the surrounding biological systems [39]. In our study, the activation, silanization, and functionalization procedures have deeply affected the roughness of titanium discs’ surfaces. The alkaline treatment with NaOH leads to a decrease in surface roughness (Ra value), from 0.882 μm in the unmodified Ti sample to 0.373 μm for the activated Ti-OH sample. These values are in complete agreement with the results obtained with SEM analyses (Figure 3). The subsequent silanization with APTES or DAPTES afforded an increase in the Ra values of 0.784 μm for Ti-APTES and 0.713 for DAPTES, thus further confirming the presence of homogeneously distributed silicon moieties (Figure 8). The functionalized samples Ti-AEMAC, Ti-GTMAC Ti-AUTEAB, and Ti-OA showed again a decrease in surface roughness with Ra values of 0.586, 0.539, 0.554, and 0.554 μm respectively, thus further confirming the effectiveness of functionalization procedures (Figure 9).

Among the different surface properties, hydrophilicity is strictly correlated with cell adhesion since cell proliferation and differentiation have been shown to increase on the surface of highly hydrophilic materials [40]. In particular, the presence of lipophilic/hydrophobic groups grafted on metal substrates has been shown to affect the interaction of metal substrates with bacteria [41]. Generally, water contact angles between 90° and 120° suggest a hydrophobic behavior, whereas smaller water contact angles are found in hydrophilic surfaces [42]. The Wenzel contact angles of all the investigated samples, measured by applying water drops on the metal surfaces are always less than 90°, thus indicating that the surfaces are hydrophilic (Figure 10). The alkaline treatment of the titanium discs with NaOH leads to a significant reduction in the surface contact angle (24.65°), compared to the untreated sample (contact angle of 71.39°). The samples functionalized with ammonium salts show different wettability behavior with increased hydrophobic character with respect to the activated sample Ti-OH. Contact angles of 56.03°, 63.02°, 88.51°, and 87.50° are observed for the Ti-AEMAC, Ti-GTMAC, Ti-AUTEAB, and Ti-OA samples, respectively. The decreased hydrophilicity observed for the Ti-AUTEAB and Ti-OA samples can be rationalized by the presence of a lipophilic alkyl chain present in both functionalized samples. The apparent static contact angle θ_w_ depends on the morphology of the surfaces and their heterogeneities which affect the contact between the liquid and the substrate.

### 3.4. Antibacterial Activity of Functionalized Titanium Discs 

The microbial biofilm represents a formation that is particularly resistant to the host immune defense and often also to antibiotic compounds [43]. The cell adhesion represents the first stage of biofilm formation and its inhibition results in a clear target for biofilm prevention [44]. Given that quaternary ammonium compounds have been shown to decrease microbial adhesion when used to coat various surfaces [19,45], in this study, a preliminary test investigating the anti-adhesion activity of functionalized titanium discs against *E. coli* (ATCC 25922) and *S. aureus* (ATCC6538) was carried out. The anti-adhesion potential of the synthesized samples Ti-AEMAC Ti-GTMAC, Ti-AUTEAB, and Ti-OA was assessed on the basis of cell viability of *E. coli* and *S. aureus* (Gram-positive) cultures, after being in contact with the films for 1.5 and 24 h, as previously reported [32]. 

Following the ultrasonic washing of the titanium discs, serial dilutions of the bacterial cultures were plated and the obtained colonies were quantified to evaluate the number of viable adhering bacteria after 1.5 and 24 h. The results, which are expressed as CFU/discs, are summarized in Figure 11 and Figure 12. For the Ti-AEMAC and Ti-GTMAC samples (*p < 0.0001*), almost superimposable results with those obtained from the starting substrate (Ti) have been recorded while a certain inhibition of bacterial adhesion was observed for Ti-AUTEAB and Ti-OA samples (*p*
*<*
*0.0001*). In particular, the Ti-AUTEAB sample showed a certain antimicrobial effect against *E coli* with inhibition of bacteria adhesion of about one log with respect to all the other samples after 1.5 h (*p*
*<*
*0.0001*) (Figure 11). In order to assess the permanence of the anti-adhesive property of the discs, the amount of the bacterial population present on the samples after 24 h of contact was evaluated (Figure 12). In line with previous studies, an increased number of adhered bacteria was observed in all the analyzed samples (*p*
*<*
*0.0001*). These experiments have shown that the functionalized titanium discs showed a similar anti-adhesive effect after one day of contact with the functionalized titanium discs to the ones obtained after 1.5 h of exposition time. These observations highlighted that the functionalized discs maintain good stability and a certain anti-adhesive effect also after 24 h.

The improved antimicrobial behavior of these samples can be rationalized considering their less hydrophilic behavior with respect to the other samples and on the basis of the chemical nature of the inserted organic moieties. Both samples possess a long alkyl chain (C_11_ for Ti-AUTEAB and C_16_ for Ti-OA) between the silicon atom and the ammonium functionality which, as also reported in studies on other cationic salts, is an important factor able to significantly increase the antibacterial activity. [46,47].

## 4. Conclusions

The possibility to develop inherent antibacterial medical devices was here investigated by covalently inserting bioactive ammonium salts onto the surface of titanium metal substrates. The bioactive molecules were anchored to titanium discs after alkaline activation and silanization procedures. The QASs AEMAC, GTMAC, and AUTEAB were linked to the metal substrates by reaction with the free nucleophilic amino group of APTES, while the carboxyl group of the naturally-derived oleic acid was used to form an ammonium salt by reaction with the dimetylamino group of DAPTES. ATR-FTIR and SEM-EDX analyses confirmed the effectiveness of the functionalization procedures. These chemical modifications were shown to deeply affect the surface properties of the metal substrates, namely roughness and hydrophilic behavior, and consequently, their interaction with their biological counterparts. The functionalized titanium discs were evaluated for their anti-adhesion properties against the Gram-negative *Escherichia Coli* and Gram-positive *Staphylococcus aureus*, at 1.5 and 24 h of bacterial contact, showing good antimicrobial activity for Ti-AUTEAB and Ti-OA samples, containing long alkyl chains between the silicon atom and the ammonium functionality. In particular, the Ti-AUTEAB sample showed inhibition of bacteria adhesion against *Escherichia Coli* of about one log with respect to the other samples after 1.5 h. Even if the functionalization with AUTEAB gave the best results in terms of inhibition of bacterial adhesion, the halogen-free Ti-OA sample could exert lower toxicity. The results of this study highlight the importance of chemical functionalization in addressing the antimicrobial activity of metal surfaces and open new perspectives in the development of inherent antibacterial medical devices. Further studies, involving the use of other bioactive molecules containing long alkyl chains as well as the use of new suitable functionalization reaction conditions will allow a better understanding of the chemical role of bioactive molecules in addressing optimal anti-bacterial adhesion properties and the evaluation of their long-term stability. 

## Figures and Tables

**Figure 1 materials-15-03283-f001:**
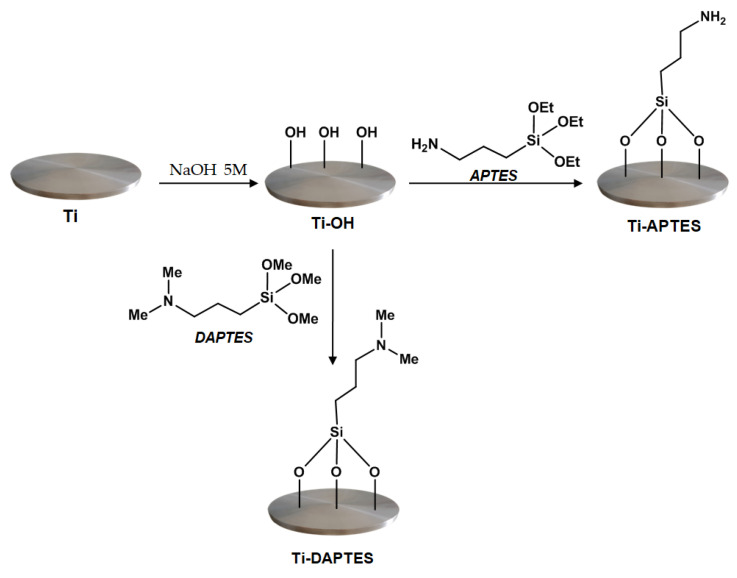
Activation and silanization of titanium discs surfaces.

**Figure 2 materials-15-03283-f002:**
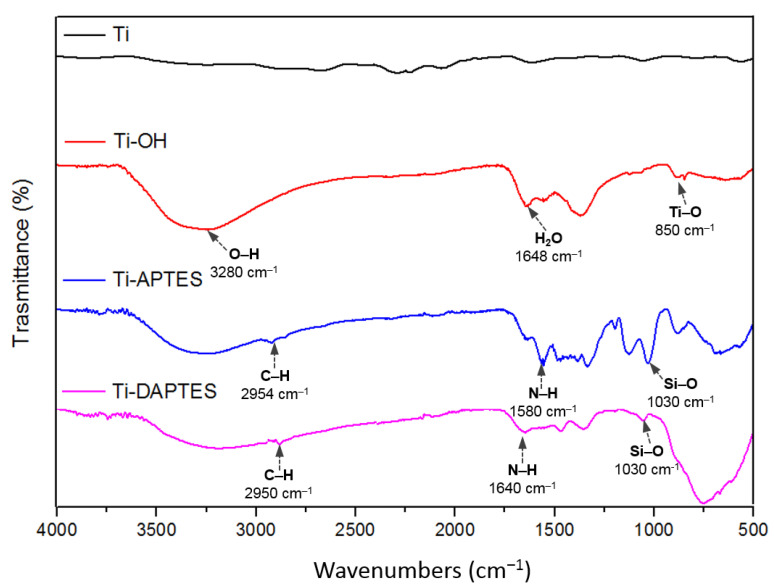
FTIR spectra of bare titanium sample (Ti), of the oxidized sample Ti-OH and of the silanized samples Ti-APTES and Ti-DAPTES.

**Figure 3 materials-15-03283-f003:**
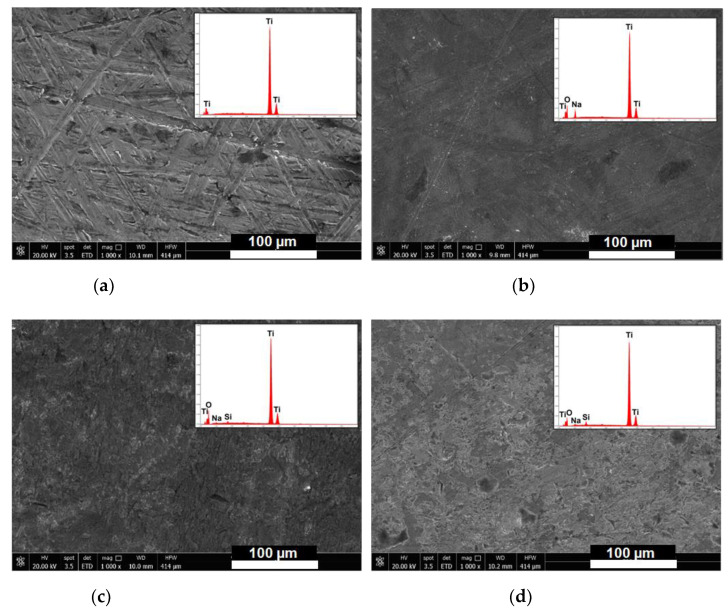
Representative Scanning Electron Microscopy/Energy Dispersive X-ray (SEM/EDX) analyses of (**a**)Ti, (**b**) Ti-OH, (**c**) TI-APTES, and (**d**) Ti-DAPTES samples.

**Figure 4 materials-15-03283-f004:**
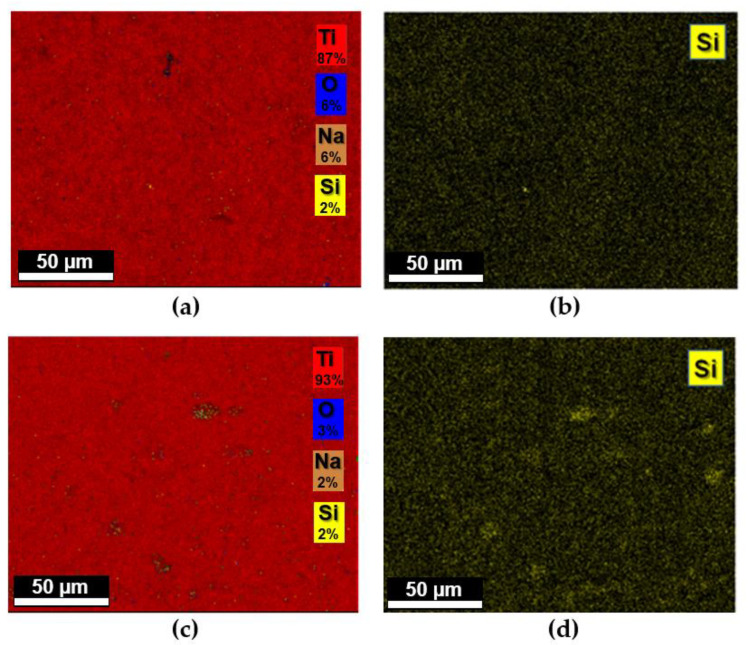
(**a**) EDX mapping analysis performed on TI-APTES; (**b**) presence of silicon atoms (yellow spots) on TI-APTES; (**c**) EDX mapping analysis performed on Ti-DAPTES; (**d**) presence of silicon atoms (yellow spots) on TI-DAPTES.

**Figure 5 materials-15-03283-f005:**
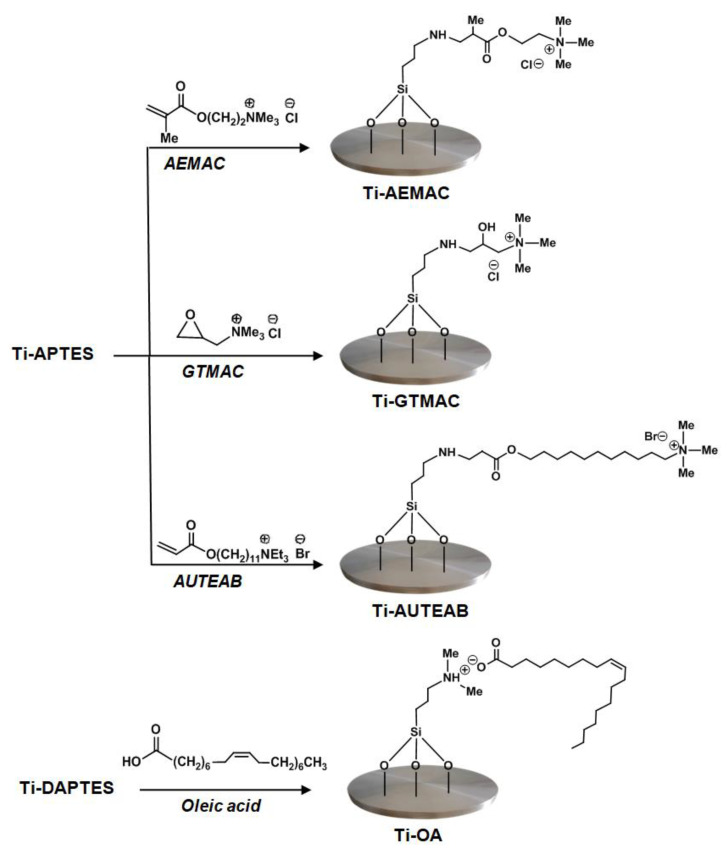
Schematic diagram illustrating the synthetic routes for titanium surface functionalization.

**Figure 6 materials-15-03283-f006:**
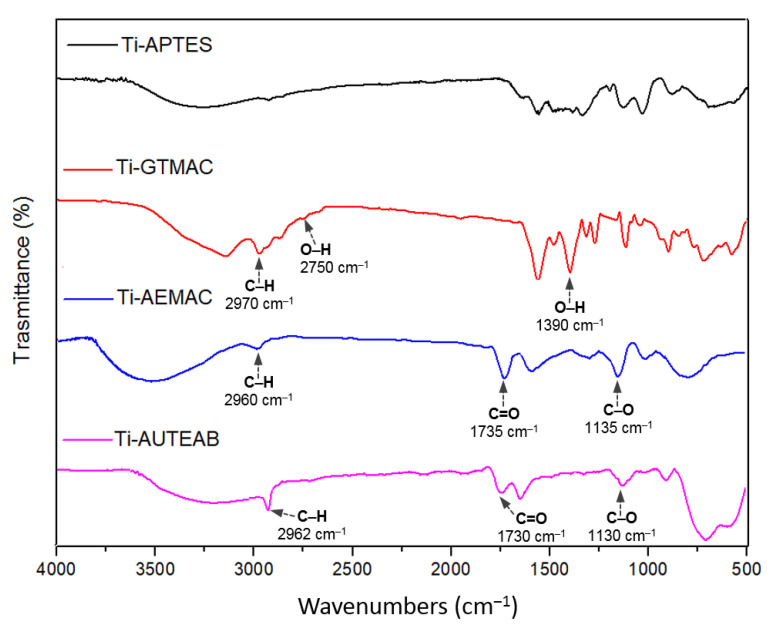
FTIR spectra of Ti-AEMAC, Ti-GTMAC, and Ti-AUTEAB samples, compared with Ti-APTES sample.

**Figure 7 materials-15-03283-f007:**
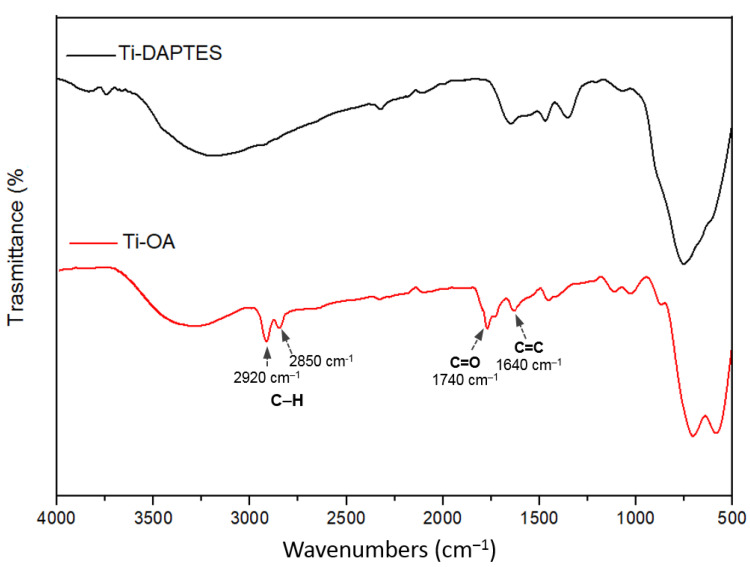
FTIR spectra of Ti-OH sample, compared with Ti-DAPTES.

**Figure 8 materials-15-03283-f008:**
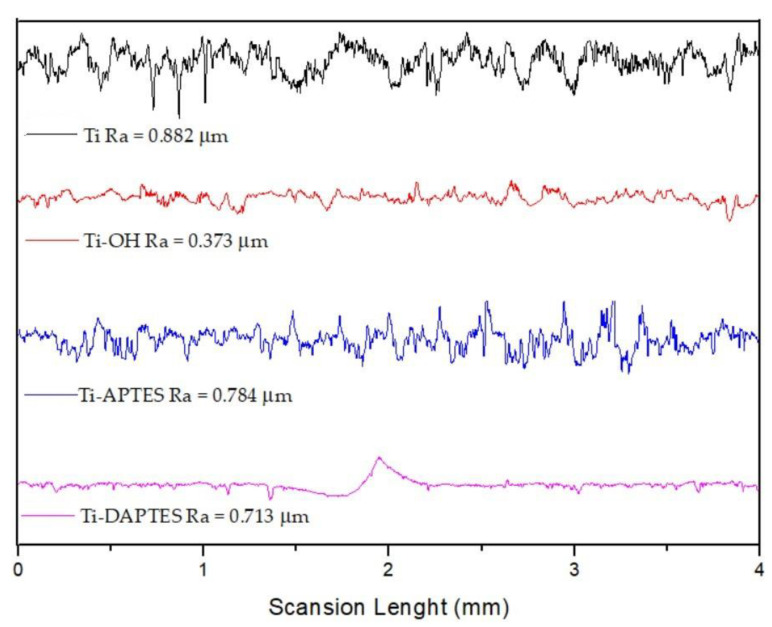
Surface roughness profiles of Ti, Ti-OH, Ti-APTES, and Ti-DAPTES samples.

**Figure 9 materials-15-03283-f009:**
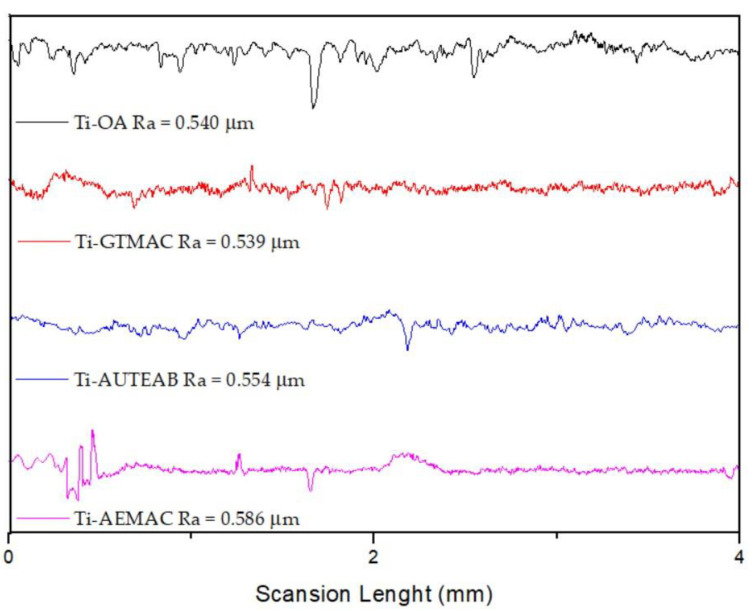
Surface roughness profiles of Ti-AEMAC, Ti-GTMAC Ti-AUTEAB, and Ti-OA, samples.

**Figure 10 materials-15-03283-f010:**
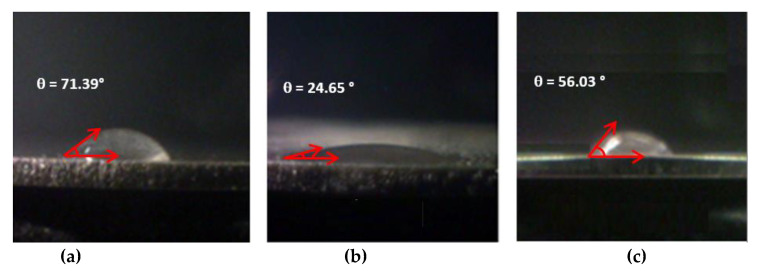
Comparison of contact angles θ_w_ and θ_Y_ of (**a**) Ti, (**b**) Ti-OH, (**c**) Ti-AEMAC, (**d**) Ti-GTMAC, (**e**) Ti-AUTEAB, and (**f**) Ti-OA samples.

**Figure 11 materials-15-03283-f011:**
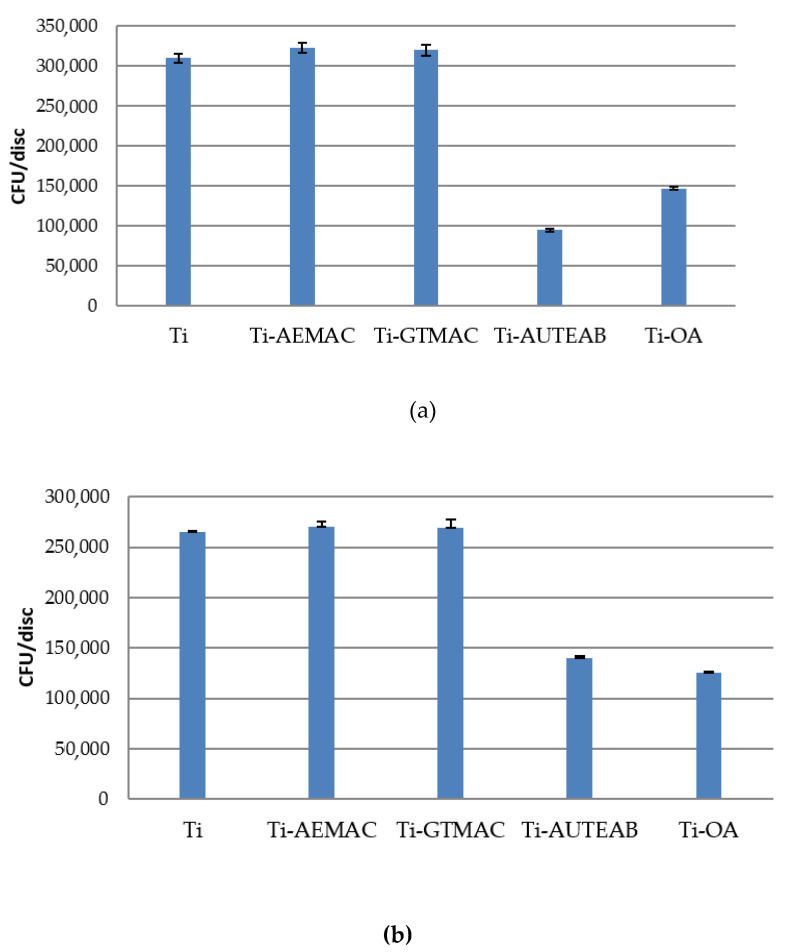
**(a**) Number of *E. coli* (**a**) and of (**b**) *S. aureus* adhered after 1.5 h of bacterial contact on bare titanium discs (Ti) and on the functionalized samples Ti-AEMAC, Ti GTMAC, Ti-AUTEAB, and Ti-OA P; Error bars indicate standard deviation (SD) from three independent CFU counting (i.e., n = 6). “Elaboration standard deviation with ANOVA, *p < 0.0001*”.

**Figure 12 materials-15-03283-f012:**
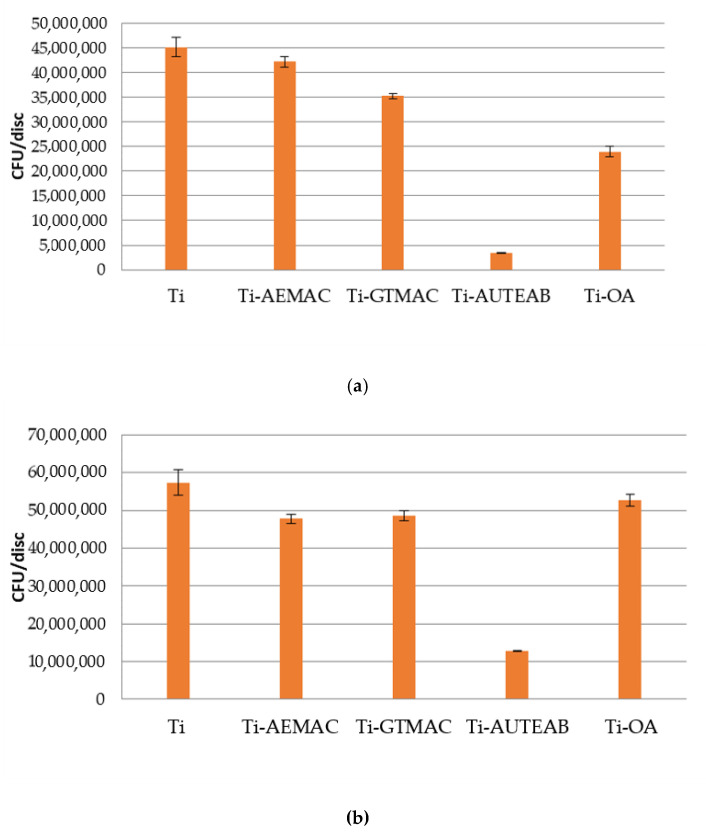
(**a**) Number of *E. coli* (**a**) and of (**b**) *S. aureus* adhered after 24 h of bacterial contact on bare titanium discs (Ti) and on the functionalized samples Ti-AEMAC, Ti GTMAC, Ti-AUTEAB, and Ti-OA P; Error bars indicate standard deviation (SD) from three independent CFU counting (i.e., n = 6). “Elaboration standard deviation with ANOVA, *p < 0.0001*”.

## Data Availability

Not applicable.

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
