# Peer review of "Titanium Surface Modification for Implantable Medical Devices with Anti-Bacterial Adhesion Properties"

_materials, 2022, doi:10.3390/ma15093283_

Round 1

Reviewer 1 Report

The manuscript by Celesti et al reports titanium surface modification for implantable medical devices with anti-bacterial adhesion properties. The manuscript presents the details and features of the methods used. The detail and clarity in the methods are good.

General comments:

  1. Manuscript needs moderate English spelling editing and grammar checks.
  2. The result and discussion part can be strengthened with more discussions and references.

Reviewer 2 Report

  • At Introduction, specify why the “N,N-dimethylaminopropyl)trimethoxysilane (DAPTES)” is not mention.
  • At section 2.1., line 120, the authors mention “all the other chemicals”. Specify the all chemicals used in this study.
  • Specify the correct name for 3-aminopropyl)triethoxysilane (APTES) and N,N-dimethylaminopropyl)trimethoxysilane (DAPTES) (lines 119, 120, 158)
  • In my opinion, I think that the sections 2.2.3-2.2.6 can be written in one paragraph.
  • At section 3.3., line 328, authors mention “This increased hydrophobicity…”. This sentence is wrong because the samples present the hydrophobic character when the contact angles is > 90°). Please, correct this paragraph.
  • Also, correct the paragraph “more hydrophobic behavior….” (line 388) because the samples don’t present contact angles > 90°)
  • The authors mention “Both samples possess a long alkyl chain (C11 for Ti-AUTEAB and C16 for Ti-OA). Explain why the contact angles are so small, because the long alkyl chains lead to growth of contact angles.

Reviewer 3 Report

This manuscript by Celesti et al. chemically modified quaternary ammonium salts on titanium discs and demonstrated its potential application in anti-bacterial adhesion devices. This work exhibits some merits while the following concerns need to be addressed before its acceptance.

1) The abstract is too long and contains too many details in the background and methods. It should be significantly shortened to focus on their quantificational anti-bacterial results.

2) Please highlight the featured absorption peaks of the samples in all FTIR images.

3) Fig. 4, the control group of EDX analysis is not included.

4) I am curious about the stability of the functionalized chemical groups on the material surfaces. Would they degrade along with time and further compromise the anti-bacterial adhesion properties? Honestly 24 h anti-bacterial adhesion tests are not sufficient for a long-term evaluation.

5) Fig. 11 and 12, any digital photographs, fluorescent or SEM images of the inhibition performance?

6) Data analysis and statistics parts are missing in the Methods. Also, please perform statistical analysis in Fig. 11 and 12.

Round 2

Reviewer 2 Report

Dear Sirs,

The manuscript was improved and it can be publish in this form.

Reviewer 3 Report

This reviewer does not have further concerns and suggests its publication.